# Evaluation of Nutritional Quality and Oxidation Stability of Fermented Edible Insects

**DOI:** 10.3390/foods14172929

**Published:** 2025-08-22

**Authors:** Anja Vehar, Doris Potočnik, Marjeta Mencin, Mojca Korošec, Blaž Ferjančič, Marta Jagodic Hudobivnik, Polona Jamnik, Ajda Ota, Lenka Kouřimská, Martin Kulma, David John Heath, Nives Ogrinc

**Affiliations:** 1Department of Environmental Sciences, Jožef Stefan Institute, 1000 Ljubljana, Slovenia; anja.vehar@ijs.si (A.V.); doris.potocnik@ijs.si (D.P.); marjeta.mencin@ijs.si (M.M.); marta.jagodic@ijs.si (M.J.H.); david.heath@ijs.si (D.J.H.); 2Jožef Stefan International Postgraduate School, 1000 Ljubljana, Slovenia; 3Department of Food Science and Technology, Biotechnical Faculty, University of Ljubljana, 1000 Ljubljana, Slovenia; mojca.korosec@bf.uni-lj.si (M.K.); blaz.ferjancic@bf.uni-lj.si (B.F.); polona.jamnik@bf.uni-lj.si (P.J.); 4Emona Nutrition Research & Development Department, Jata Emona, 1000 Ljubljana, Slovenia; ajda.ota@e-rcp.si; 5Department of Microbiology, Nutrition and Dietetics, Czech University of Life Sciences, 165 00 Praha-Suchdol, Czech Republic; kourimska@af.czu.cz; 6Department of Zoology and Fisheries, Czech University of Life Sciences, 165 00 Praha-Suchdol, Czech Republic; kulma@af.czu.cz

**Keywords:** edible insects, fermentation, nutritional value, amino acids, fatty acids, elemental composition, oxidation stability

## Abstract

Fermentation, a traditional method for enhancing nutritional value and functionality, has significant potential for improving the quality, safety and acceptability of farmed insect products. In this study, yellow mealworm, house cricket and migratory locust were fermented using *Lactobacillus plantarum* and a commercial starter culture for 48 h. Samples were analyzed for proximate composition, amino and fatty acid profiles, elemental composition and oxidation stability. Fermentation reduced total dietary fiber in yellow mealworm (33%) and house cricket (12%), and increased non-protein nitrogen (38% and 16%), while total and protein nitrogen remained unaffected. Fatty acid profiles also remained unchanged, whereas the amino acid composition varied depending on the species and fermentation culture. Essential mineral concentrations varied depending on species and fermentation culture Fe (19–23%), K (25%), Mg (12–23%), Mn (36–378%), Na (20–49%) and P (22%) increased, levels of Se (15%), and Cu (16%) decreased, while Zn levels showed inconsistent trends among treatments. Oxidation stability of yellow mealworm (41–42%) and migratory locust (21–29%) decreased, but improved for house cricket (153–167%). Overall, fermentation enhanced the nutritional value of edible insects, although the extent of improvement varied by species and fermentation culture.

## 1. Introduction

Edible insects represent a diverse and nutrient-rich source of macro- and micronutrients, increasingly recognized as a sustainable alternative to traditional animal-based protein [1]. Compared to conventional livestock, farmed insects have a significantly lower environmental footprint. For example, they require 75–95% less water and 80–90% less land. They also emit up to 80% fewer greenhouse gases while providing high-quality protein with a complete amino acid profile, essential fatty acids, and key micronutrients such as copper, iron, magnesium, zinc, and B vitamins [2]. In addition, insect farming is adaptable to vertical systems, generates minimal noise, and produces almost no waste, as insect frass can be used as an organic fertilizer [3,4]. Moreover, utilizing agricultural by-products and food waste as insect feed can further reduce production costs, addressing a major barrier to market entry [5,6,7,8,9], while contributing to food waste reduction in regions where such practices are allowed under local regulations [10,11,12,13,14].

The European Commission also recognizes the potential of insects as sustainable sources of food and feed by approving several species as novel foods under various regulations. Approved species include yellow mealworm (*Tenebrio molitor*; Commission Regulation 2021/882, 2022/169, 2025/89) [15,16,17], migratory locust (*Locusta migratoria*; Commission Regulation 2021/1975) [18], house cricket (*Acheta domesticus*; Commission Regulations 2022/188, 2023/5) [19,20], and lesser mealworm (*Alphitobius diaperinus*; Commission Regulation 2023/58) [21]. Furthermore, Commission Regulation 2021/1372 authorizes eight insect species for feed in poultry, pig, and fish farming [22]. These include tropical house cricket (*Gryllodes sigillatus*), black soldier fly (*Hermetia illucens*), Jamaican field cricket (*Gryllus assimilis*), house fly (*Musca domestica*), and silkworm moth (*Bombyx mori*), in addition to the aforementioned species.

Although insects have a long history of widespread consumption in several countries [23], they are met with significant skepticism, safety concerns, and unfamiliarity in Western societies [24,25]. One of the main factors influencing consumer perception is related to product characteristics, such as taste, price, texture, appearance, smell, the degree of insect visibility, and the proportion of insect-based ingredients [24]. However, these characteristics can be changed and improved using differing processing techniques, such as blanching, boiling, curing, drying, fermentation, freeze-drying, deep frying, marination, roasting, smoking and stewing, which can also reduce microbial loads and extend insect product shelf life [8,26,27,28,29].

Among these methods, fermentation is a widely recognized technique for improving the nutritional, functional, and sensory qualities of various food products [30]. The process involves spontaneous or controlled (starter culture-assisted) microbial activity that breaks down large molecules into smaller ones through various oxidation–reduction reactions. It can serve multiple purposes, including (i) enhancing sensory qualities such as flavor, aroma, and texture; (ii) increasing nutritional value by improving nutrient bioavailability; (iii) eliminating anti-nutritional factors; (iv) increasing levels of bioactive compounds, such as amino acids, polyphenols, organic acids, (v) promoting intestinal health by incorporating beneficial microorganism; and (vi) extending shelf life and reducing microbiological risks by inhibiting pathogenic microorganisms [30,31,32,33].

Recent studies have begun to explore the potential of fermentation for improving the nutritional and functional properties of edible insects, particularly species such as yellow mealworm [34,35,36,37], black soldier fly [38,39,40,41], house cricket [42,43], desert locust [36] and Mexican grasshopper [37]. Various microbial cultures, primarily lactic acid bacteria, *Bacillus* spp., and *Aspergillus* spp., have been used, with reported improvements in protein digestibility, amino acid profiles, fatty acid composition, antioxidant activity, shelf life and microbiological safety.

Changes in proximate composition and nutrient metabolism appear consistently across species, although the underlying mechanisms differ. For example, lactic acid fermentation of black soldier fly larvae and related products has been shown to modify proximate composition, amino acid and fatty acid profiles, reduce chitin content, and show antimicrobial activity against pathogens [38]. Also, similar studies using *Bacillus subtilis*, *Aspergillus niger*, *Lactobacillus crispatus* and *Pichia kudriavzevii*, report shifts in carbon, amino acid, and fatty acid metabolism as well as changes in bioactive components and microbial community structure [39,40,41]. In house cricket, *Lactobacillus plantarum* fermentation has been reported to modify proximate composition, organic acids, alcohols, fatty and amino acid profiles, volatile compounds, mineral content, and the composition of the microbiome [43,44]. Other approaches, including the use of methods that mimic Thai fermented shrimp paste, have highlighted effects on peptide content, mineral composition, amino acids, and in vitro protein digestibility [42]. In yellow mealworm, fermentation with lactic acid bacteria, commercial meat starter cultures, and *Aspergillus oryzae* has been reported to modify proximate composition, amino acid and fatty acid profiles, antioxidant activity, and microbiome composition [44], along with microbial counts, sugar consumption, organic acid production, and sensory properties [26,34,35,36,37].

While these studies demonstrate the potential of fermentation to improve the nutritional and functional quality of edible insects, variations in insect species, fermentation conditions, and analytical approaches limit direct cross-study comparisons. Moreover, there remains a substantial knowledge gap regarding how different fermentation cultures and conditions influence the nutritional, functional, and safety characteristics of various insect species.

The present study investigates the effects of fermentation using *Lactobacillus plantarum* and a commercial starter culture on three farmed edible insect species: yellow mealworm, migratory locust, and house cricket, in terms of proximate composition, amino acid and fatty acid profiles, elemental composition, and oxidation stability. This work addresses an existing knowledge gap, as data on yellow mealworm and house cricket under such fermentation conditions remain limited, and, to the authors’ knowledge, no studies on fermented migratory locust have yet been reported.

## 2. Materials and Methods

### 2.1. Chemicals and Reagents

Dichloromethane (CH_2_Cl_2_), methanol (MeOH), and hexane (C_6_H_14_) for organic residue analysis were sourced from J. T. Baker B.V. (Deventer, The Netherlands). Sodium hydroxide (NaOH) (puriss. p.a., ACS reagent, ≥98%) and 14% boron trifluoride in methanol were obtained from Sigma-Aldrich (St. Louis, MO, USA). Trichloroacetic acid (≥99.5%) was obtained from Merck, Germany. Ultrapure water (18.2 MΩ) was prepared using the Milli-Q System (Merck Millipore, Watertown, MA, USA). A Supelco 37 Component FAME Mix in dichloromethane, containing methyl esters of 37 fatty acids, was obtained from Supelco (Bellefonte, PA, USA). Amino acid standards (≥98%) used for analysis included L-valine, L-leucine, L-isoleucine, L-threonine, L-serine, L-proline, L-methionine, L-glutamic acid, L-phenylalanine, L-lysine, L-histidine, and L-tyrosine, sourced from Sigma Aldrich. Standards for alanine, glycine, and aspartic acid were purchased from Fluorochem (Derbyshire, UK). The internal standard, Stable Isotope Labelled Amino Acid Mix Solution 1, containing labelled variants of 17 amino acids, was obtained from Supelco (Buchs, Switzerland). Hydrochloric acid (37%) was sourced from VWR Chemicals (Fontenay-sous-Bois, France), while thioglycolic acid (≥99%, France) and phenol (≥99%, USA) were purchased from Sigma Aldrich. All other reagents for amino acid analysis were provided in the EZ Amino Acid Analysis Kit from Phenomenex (Torrance, CA, USA). For elemental composition analysis, nitric acid (65% Suprapur^®^ for trace analysis, HNO_3_) and hydrogen peroxide (30% Suprapur, for trace analysis, H_2_O_2_) were obtained from Supelco. For insect fermentation, De Man, Rogosa and Sharpe (MRS) broth was obtained from Merck (Darmstadt, Germany), and sodium chloride (NaCl) was obtained from Sigma-Aldrich (Steinheim, Germany).

### 2.2. Insect Rearing

Three edible insect species, the yellow mealworm, the house cricket, and the migratory locust, were selected based on their approval as novel foods by the EU (Commission Regulations 2021/882, 2021/1975, 2022/169, 2022/188, 2025/89). All species were sourced from the insect-rearing facility at the Faculty of Agrobiology, Food, and Natural Resources, Czech University of Life Sciences, Prague and were maintained under controlled conditions.

House crickets and mealworms were reared in a rack system under controlled conditions at 27 ± 1 °C, 40–50% relative humidity, and a 12:12 h light–dark photoperiod. Migratory locusts were maintained at the same temperature and photoperiod, but at a lower relative humidity of 30%. The humidity was controlled using a DeLonghi dehumidifier (Treviso, Italy) and measured daily with a Comet C3120 thermohygrometer (Comet system, Rožnov pod Radhoštěm, Czech Republic). No additional heating was used for house crickets and yellow mealworms, while migratory locusts were provided with heating foils and 40 W bulbs to create warm zones. House crickets and migratory locusts were reared in plastic boxes (56 × 39 × 28 cm, IKEA, Prague, Czech Republic), fitted with egg trays (Schubert Partner, Prague, Czech Republic) and secured with 80% aluminum anti-insect mesh lids. Yellow mealworms were reared in 60 × 40 × 12.5 cm breeding trays (Beekenkamp, Maasdijk, The Netherlands).

Insects were fed *ad libitum*. House crickets received chicken feed (77.9% wheat, 17.6% soybean meal, 1.8% rapeseed oil, 2.7% mineral and vitamin premix; particle size < 1 mm), while migratory locusts were provided wheat bran, grass hay, and fresh Poaceae grass. Mealworms were fed on a 4:1 mixture of wheat bran and chicken feed. Water gel (Oslavan, Náměšť nad Oslavou, Czech Republic) was supplied daily for house crickets and migratory locusts. House crickets and locusts were maintained in the same containers from hatching until harvest at the adult stage, while mealworms were harvested upon first pupal emergence.

All insects were starved for 24 h prior to euthanasia by freeze-killing (−20 °C for 24 h). Samples were lyophilized (Trigon Plus, Čestlice, Czech Republic), homogenized (A10; IKA Werke GmbH & Co. KG, Staufen, Germany), and stored at −20 °C until further analysis. The dry matter content (%) of fresh insects for further calculations was determined by calculating the mass difference before and after lyophilization and was 30%.

### 2.3. Insect Fermentation

The insect samples (yellow mealworm, house cricket and migratory locust) were subjected to lactic acid fermentation using *L. plantarum* and a commercial starter culture for meat processing. *Lactobacillus plantarum* IM 527 was obtained from the Institute of Dairy Science and Probiotics, Department of Animal Science, Biotechnical Faculty, University of Ljubljana, Slovenia. An aliquot of *L. plantarum* stock culture (100 μL) was inoculated into 20 mL of sterilized liquid MRS medium and incubated on a rotary shaker (150 rpm, 30 °C, 12 h) until the late exponential phase was reached. The obtained broth (2 mL) was then centrifuged (10,000× *g*, 5 min) and washed twice with a saline solution to obtain the inoculum suspension.

A stock suspension of a commercial starter culture SM-194 Bactoferm^®^ (Chr Hansen A/S, Hørsholm, Denmark) was prepared according to the manufacturer’s instructions by dissolving 0.125 g of the culture in 5 mL of sterile saline solution. The starter culture comprised *Staphylococcus xylosus*, *Debaryomyces hansenii*, *Lactobacillus sakei*, *Staphylococcus carnosus*, and *Pediococcus pentosaceus*.

Both fermentation types were performed using 8 g of lyophilized and milled insect material, placed in 50 mL polyethylene centrifuge tubes and then capped. For house cricket and migratory locust, a substrate-to-sterile double distilled H_2_O ratio of 1:2 (8 g:16 mL) was used, whereas a ratio of 1:1.5 (8 g:12 mL) was applied for yellow mealworm. Samples were then inoculated with the *L. plantarum* or commercial starter culture suspension (240 μL for house cricket and migratory locust; 200 μL for yellow mealworm) and incubated under static conditions at 30 °C for 48 h.

Fermentation was performed in triplicate. To monitor the fermentation process, samples were collected at 0, 24, and 48 h, and the pH and colony-forming units (CFU) were measured. A reduction in pH (3–30%), accompanied by an increase in CFU count (29–47%) over the fermentation period, indicated active microbial growth and metabolic activity, confirming successful fermentation. Following fermentation, all samples were lyophilized (72 h, −53 °C, 0.280 mbar) and stored at −20 °C until further analysis.

### 2.4. Determination of Proximate Composition

Proximate composition of yellow mealworm and house cricket, both unfermented and fermented with *L. plantarum*, was determined using official AOAC methods for macronutrient analysis, though due to insufficient sample amounts, the analysis was limited to these two species. This decision was based on several practical considerations: (1) proximate composition requires relatively large sample quantities compared to other analyses in this study; (2) initial difficulties in rearing migratory locust resulted in limited biomass; and (3) the insectarium’s capacity to produce large-scale insect biomass was constrained. Consequently, yellow mealworm and house cricket were selected as model species due to their current market relevance, relative ease of rearing, and established production protocols. *Lactobacillus plantarum* was chosen for fermentation as it is one of the most extensively studied lactic acid bacteria, particularly in the context of food fermentation.

The moisture content was measured following AOAC 950.46 [45], and dry matter was calculated as 1—the moisture content. Ash, total fat, and protein contents were determined using AOAC 920.153, 991.36, and 928.08 [45], respectively. Protein content was calculated from nitrogen (Kjeldahl method) using a factor of 6.25, after subtracting non-protein nitrogen (NPN).

The NPN extracts were prepared using a modified DeVries et al. (2017) method [46,47]. A known amount (2.00 g) of sample was mixed with 10 mL distilled water and 10 mL 24% trichloroacetic acid, homogenized (Ultra-Turrax, 13,500 rpm, 30 s), sonicated (10 min), centrifuged (4000 rpm, 10 min), and filtered with filter paper discs (grade 389, 125 mm diameter, 8–12 µm pore size, 100% cotton linters, Sartorius, Germany). NPN was determined from the clear filtrate (5 g) by the Kjeldahl method.

Dietary fiber content was determined using the enzymatic–gravimetric AOAC 991.43 method [45], employing heat-stable α-amylase, protease, and amyloglucosidase (Merck KGaA, Germany). Insoluble dietary fiber was recovered by filtration, while soluble dietary fiber was precipitated from filtrates with 96% ethanol. Ash and protein contents of residues were corrected, and total dietary fiber was the sum of insoluble and soluble fractions. Available carbohydrate content (g/100 g) was calculated using Equation (1) [48]:(1)Available carbohydrate=100 g/100 g−(water+ash+protein+fat+fiber)
where protein is calculated as total nitrogen × 6.25, thereby accounting for both protein and non-protein nitrogen fractions.

The energy value (EV, kJ/100 g) was calculated as the sum of the energy values for each macronutrient using the following equation [48]:(2)$$\mathrm{EV}={\mathrm{EV}}_{\mathrm{protein}}+{\mathrm{EV}}_{\mathrm{fat}}+{\mathrm{EV}}_{\mathrm{available carbohydrates}}+{\mathrm{EV}}_{\mathrm{dietary fibre}}$$

The energy value of each nutrient was calculated by multiplying its content (g/100 g) by the corresponding energy conversion factor: 17 kJ/g for protein and available carbohydrates, 37 kJ/g for fat, and 8 kJ/g for dietary fiber [48].

### 2.5. Fatty Acid Analysis

Freeze-dried insect samples (200 mg) were weighed into screw-cap vials. To extract total lipids, 500 μL of dichloromethane and 3 mL of 0.5 M sodium hydroxide in methanol were added. The samples were purged with nitrogen gas, heated for 10 min at 90 °C, and then rapidly cooled in an ice bath. After cooling, 3 mL of boron trifluoride in methanol was introduced to convert the lipids into fatty acid methyl esters (FAMEs). The extract was then purged with nitrogen and heated for 10 min at 90 °C. Once cooled, the FAMEs were extracted using 3 mL Milli-Q water and 1.5 mL hexane. The organic phase was transferred to a GC vial and stored at −20 °C. All samples were prepared in duplicate.

All samples were analyzed using an Agilent 6890N gas chromatograph (Network GC System, Agilent Technology, Santa Clara, CA, USA) with a flame ionization detection (FID). Separation was achieved on a CP-Sil 88 capillary column (100 m × 0.25 mm × 0.20 µm, Agilent J&W). The temperature program was as follows: 100 °C (5 min), ramped to 180 °C at 8 °C/min (9 min), and then increased to 230 °C at 1 °C/min (1 min). Helium was used as the carrier gas in constant flow mode (1.3 mL/min), with an injection volume of 1 μL and a split ratio of 10:1. The inlet and detector were set to 260 °C and 250 °C, respectively.

Fatty acids were identified and quantified by comparing their retention times with the Supelco 37-component FAME mix standard. The results were expressed as relative weight percentages of total fatty acids. Procedural blanks were analyzed alongside each batch of samples, and the FAME mix standard was run after every ten samples to verify system stability.

Fatty acids were categorized as saturated fatty acids (SFA), monounsaturated fatty acids (MUFA), and polyunsaturated fatty acids (PUFA), with a specific focus on n − 3 and n − 6 fatty acids. The PUFA/SFA and n − 6/n − 3 ratios were also calculated. Additionally, the atherogenic index (IA), thrombogenic index (IT), and hypocholesterolemic/hypercholesterolemic ratio (h/H) for the total lipid content were calculated according to the following equations [49]:(3)$$IA=\frac{C12:0+(4\times C14:0)+C16:0}{\Sigma MUFA+\Sigma n-6+\Sigma n-3}$$(4)$$IT=\frac{C14:0+C16:0+C18:0}{\left(0.5\times \Sigma MUFA)+(0.5\times \Sigma n-6)+(3\times \Sigma n-3)+(n-3/n-6\right)}$$(5)$$h/H=\frac{C18:1+\Sigma PUFA}{C12:0+C14:0+C16:0}$$

### 2.6. Amino Acid Analysis

A known weight of lyophilized insect sample (50 mg) was placed into hydrolysis micro-reaction vessels, and 2.5 mL of 4% thioglycolic acid in 6M HCl and 25 μL of liquid phenol (heated to 80 °C) were added. The mixture was then vortexed. The samples were then purged with nitrogen for 5 min, sealed, and heated at 110 °C for 24 h. During hydrolysis, glutamine was converted to glutamic acid, and asparagine to aspartic acid; therefore, reported values for glutamic and aspartic acids represent the combined total of these amino acids. After hydrolysis, the samples were filtered through CHROMAFIL Xtra PTFE 0.45 µm syringe filters (Macherey-Nagel, Düren, Germany).

Amino acid analysis was performed using the EZ:faastTM kit. First, 50 μL of the filtered hydrolysate was transferred into a sample preparation vial, followed by the addition of 100 μL of 1 M Na_2_CO_3_ solution and vortexing. A 30 μL aliquot of the pH-adjusted mixture (pH 2) was then transferred into a new vial, and 50 μL of internal standard (200 nmol/mL) was added, followed by vortexing. The sample was then passed through a sorbent tip using a 1.5 mL syringe. After adding 200 μL of Milli-Q water into the vial, the sample was passed through the sorbent tip again, and the liquid was discarded. The sorbent tip was then detached.

The compounds of interest were eluted using 200 μL of a NaOH/N-propanol/3-picoline mixture (3/1.6/0.4, *v*/*v*/*v*). This elution was performed by attaching the sorbent tip to a 0.6 mL syringe, pulling back the plunger halfway, and wetting the sorbent with the eluting medium before expelling the liquid into the sample vial. A 50 μL aliquot of propyl chloroformate/chloroform/iso-octane (1:3:1, *v*/*v*/*v*) was added, and the contents were emulsified by vortexing for 5 s. The reaction was allowed to proceed for 1 min and repeated once more. After this, 100 μL of a 9:1 (*v*/*v*) iso-octane/chloroform mixture was added and vortexed for 1 min. One hundred μL of the organic layer was transferred into a GC autosampler vial, evaporated under nitrogen for 10 min, and reconstituted in 100 μL iso-octane/chloroform (4:1, *v*/*v*). The reconstituted sample was transferred into a GC-MS vial with a glass insert and analyzed. All samples were prepared in duplicate.

Samples were analyzed on a 7890B series GC coupled to a 5977A single quadrupole mass selective detector (Agilent, USA). Separation was achieved using a ZB-AAA GC column (10 m × 0.25 mm, Phenomenex, USA), with helium as the carrier gas (1.1 mL/min). A volume of 1.5 μL of the sample was injected in split mode (1:15). The GC oven temperature program was 110 °C (1 min), then ramped at 30 °C/min to 320 °C (total runtime: 7 min). The inlet and transfer line were set to 250 °C and 260 °C, respectively. The mass spectrometer was operated in electron impact (EI) mode at 70 eV, with a source temperature of 240 °C, a quadrupole temperature of 180 °C, and an auxiliary temperature of 310 °C. The scan range was 45–440 m/z, and the compounds were identified and quantified using selected ion monitoring (SIM) mode (Appendix A). The data were processed using MassHunter software (version 10.0., Agilent Technologies, USA).

The ratio of essential to total amino acids (E/T, %) and the essential amino acid index (EAAI) were determined according to FAO (2013) [50] based on valine, leucine, isoleucine, threonine, phenylalanine + tyrosine, methionine, histidine, and lysine. Cystine and tryptophan were excluded from the calculations due to method limitations, since acid hydrolysis degrades tryptophan and cystine. The formula for EAAI is as follows:(6)$$\mathrm{EAAI}=\sqrt[{8}]{\frac{g\mathrm{of}\mathrm{lysine}\mathrm{in}100\;g\mathrm{of}\mathrm{analysed}\mathrm{protein}}{g\mathrm{of}\mathrm{lysine}\mathrm{in}100\;g\mathrm{of}\mathrm{reference}\mathrm{protein}}\times (\mathrm{etc}.\mathrm{for}\mathrm{other}\;7\mathrm{EAA})}\times 100$$

The index is based on the sum of all amino acids compared to those of the reference protein for older children (≥3 years) and adults [50]. The amino acid with the lowest amino acid score (AAS) is considered the limiting amino acid. The AAS was calculated according to the following equation.(7)$$\mathrm{AAS}=\frac{\mathrm{mg}\mathrm{amino}\mathrm{acid}\mathrm{per}\;g\;\mathrm{of}\mathrm{protein}}{\mathrm{mg}\mathrm{amino}\mathrm{acid}\mathrm{per}\;g\;\mathrm{of}\mathrm{reference}\mathrm{protein}}\times 100$$

### 2.7. Elemental Analysis

Sample digestion for elemental analysis was performed following Vehar et al. (2025) [51]. Briefly, a known quantity of lyophilized insect sample (0.10–0.15 g) was placed into a pre-cleaned Teflon digestion vial. Subsequently, 1 mL of HNO_3_ and 250 µL of H_2_O_2_ were added to each vial. All samples were microwave-digested using an Ultrawave Single Reaction Chamber Microwave Digestion System (Milestone, Italy). The digestion program involved heating the samples to 240 °C over 20 min, followed by a 15-min hold at 100 bars with a maximum power of 1500 W. After digestion, the digestate was quantitatively transferred to a polyethylene-graduated tube and diluted to a final volume of 10 mL with Milli-Q water. For measuring macro elements, samples were further diluted with 5% HNO_3_.

The determination of selected elements (Ag, Al, As, B, Ba, Ca, Cd, Co, Cs, Cu, Fe, Hg, K, Mg, Mn, Mo, Na, Ni, P, Pb, Rb, S, Sb, Se, Sn, Sr, U, V, and Zn) was conducted using an Agilent 8800 triple quadrupole inductively coupled plasma mass spectrometer, QQQ-ICP-MS (Agilent, Japan, Tokyo). Different reaction gases, H_2_, He and O_2_ were used. Internal standards (Y, Rh, Sc, and Gd) were added online in all modes, and external calibration was used to quantify the results. All samples were prepared in duplicate. Additionally, blanks and certified reference materials (SRM NIST 1566b Oyster tissue, Typical Diet NIST 1548b) were prepared in each run in the same manner to ensure the quality and accuracy of the results. The limit of detection (LOD) was calculated as three times the standard deviation of the blank samples.

A risk assessment for toxic elements was performed based on an adult (70 kg) ingesting 100 g of fresh insects. The intake of each element was first calculated (Appendix A) and then compared with reference values (Appendix A) obtained from OpenFoodTox: EFSA’s chemical hazards database [52]. Based on the concentrations of essential elements and their respective reference daily intakes (RDI; Appendix A), the percentage contribution to daily intake for adults was reported, based on a consumption scenario of 100 g of fresh insects per day.

### 2.8. Oxidation Stability Analysis

To assess the oxidation stability, a known amount (3.5 g) of lyophilized sample was placed in the test chamber of a RapidOxy 100 Oxidation Stability Tester (Anton Paar, Austria). The induction time was measured at 120 °C and 700 kPa, with continuous monitoring of oxygen consumption. The induction time was recorded when the pressure drop reached 10%, indicating oxidation stability. The measurement error is 2–3%.

### 2.9. Statistical Analysis

Data processing and visualization were performed using R-based software (version 4.2.3). All data sets were checked for normal distribution using the Shapiro-Wilk test and for homoscedasticity using Levene’s test. For proximate composition, where only yellow mealworm and house cricket before and after fermentation with *L. plantarum* were analyzed, significant nutritional parameters were assessed using pairwise comparisons. Depending on the assumptions for parametric testing (normality and homoscedasticity of variance), either a *t*-test or the Mann-Whitney *U* test was applied. If the assumptions were not met, the Mann-Whitney *U* test was used; otherwise, the *t*-test was applied. For all other analyses, where three insect species were evaluated before and after fermentation with *L. plantarum* and a commercial starter culture, significant nutritional and compositional parameters were assessed using one-way ANOVA or the Kruskal-Wallis test, depending on whether the assumptions for parametric testing were satisfied. When assumptions were violated, the Kruskal-Wallis test was applied; otherwise, one-way ANOVA was used. The statistical significance level was 0.05 with *p* ≤ 0.05, indicating rejection of the null hypothesis. Fermented and unfermented samples from the same species were grouped to ensure reliable statistical comparisons between species. However, due to the insufficient number of samples, a statistical comparison between fermented and unfermented samples was not possible.

## 3. Results and Discussion

### 3.1. Proximate Composition

Due to limited sample availability, the proximate composition was assessed only for yellow mealworm and house cricket before and after 48 h fermentation with *L. plantarum*. Proximate composition in terms of ash, total fat and proteins (Table 1) was comparable to those obtained by Cho et al. (2018), Khatun et al. (2021), Kulma et al. (2019), Messina et al. (2019) and Yi et al. (2013), while Kittibunchakul, Whanmek, and Santivarangkna (2023) and Vasilica et al. (2022) report higher values for total protein and lower values for total fat [35,42,43,53,54,55,56]. Statistical analyses (Appendix A) showed that yellow mealworm had statistically significantly higher (*p* < 0.05) ash concentrations (4.35 to 4.50 g/100 g) and lower total nitrogen content (8.95 to 9.15 g/100 g) compared to house cricket (ash: 4.23 to 4.30 g/100 g, total nitrogen: 9.24 to 9.30 g/100 g).

The results show a decrease in total dietary fiber content following fermentation in both insect species (33% for yellow mealworm and 12% for house cricket), primarily due to a reduction in insoluble dietary fiber (38% and 16%, respectively). Although the values for soluble dietary fiber are very low and therefore imprecise, the measurement showed a slight increase in the yellow mealworm (from 0.025 g/100 g to 0.366 g/100 g). The increase in soluble fiber after fermentation with *L. plantarum* has been previously reported in other commodities, as the enzymatic breakdown of insoluble fiber during fermentation leads to its conversion into soluble forms [57,58,59].

Regarding nitrogen composition, total nitrogen and protein nitrogen remained unchanged in yellow mealworm and house cricket, while non-protein nitrogen increased (221% and 159%, respectively). Despite the rise in non-protein nitrogen, the total nitrogen and protein nitrogen contents stayed constant, suggesting that microbial activity was breaking down proteins, but not enough to reduce the protein content in the insects significantly. Instead, the nitrogen is likely redistributed into various forms, such as amino acids and peptides, maintaining the overall nitrogen balance. This redistribution has already been shown, for example, in *Broussonetia papyrifera* fermented using *L. plantarum* during ensiling [59].

Following fermentation, only yellow mealworm exhibited an increase in available carbohydrates (from 0.98 g/100 g to 6.71 g/100 g), while no significant differences were observed in protein content, energy value, or total fat for either species. The significant increase in available carbohydrates may be due to the fermentation process breaking down complex polysaccharides and fibers into simpler sugars, which are then released as available carbohydrates [60]. The lack of significant changes in protein content, energy value, or total fat suggests the limited impact of *L. plantarum* on protein and fat molecules, as these macronutrients are more resistant to microbial degradation compared to carbohydrates and fibers. Additionally, the fermentation conditions might not have been optimized for significant lipid breakdown or protein denaturation, which can require more specialized conditions or longer fermentation periods.

Comparison with existing literature on edible insects is complicated by the variability in experimental design, insect species, fermentation culture, and analytical methodology in the limited number of available studies. For instance, Cho et al. (2018) fermented raw and defatted yellow mealworm using a soy sauce fermentation process involving *Aspergillus oryzae* and *Bacillus licheniformis*, reporting increased total and protein nitrogen levels after six days, which remained stable over the 13- and 20-day fermentation period [35]. Liu et al. (2022) fermented black soldier fly larvae for four days with *Bacillus subtilis* and *Aspergillus niger*, observing a reduction in actual protein (40%) and fat (30%) content [39]. In the study by Kittibunchakul et al. (2023), house cricket was fermented using a traditional Thai shrimp paste method, employing Kapi as a starter culture, which predominantly contains lactic acid bacteria [42]. After 4 weeks of fermentation, they noted significant reductions in energy (24%), protein (23%), fat (21%), and carbohydrates (43%). Meng et al. (2023), who fermented black soldier fly larvae for 24 h using *Lactobacillus crispatus* and *Pichia kudriavzevii* cultures, also observed a decrease in crude fat content, while crude protein remained unchanged [40]. Also, Pérez-Rodríguez, Ibarra-Herrera, and Pérez-Carrillo (2023) fermented yellow mealworm and Mexican grasshopper with *Aspergillus oryzae*, finding that after four days, protein content decreased (49%, 16%, respectively) in both species, while fat content increased (14%, 119%, respectively), and crude fiber remained unaffected [37].

### 3.2. Fatty Acid Composition

A total of seven FAs were detected in amounts >1% in the studied insects (Table 2), including three SFAs (myristic, palmitic, stearic), two MUFAs (palmitoleic and oleic acid) and two PUFAs (linoleic—LA and α-linoleic acid—ALA). Their relative levels (>1%) for yellow mealworm were as follows: oleic > LA > palmitic > myristic > stearic > palmitoleic > ALA. The relative levels for house cricket were: oleic > LA > palmitic > stearic > ALA > palmitoleic > myristic acid, while for migratory locust: LA > oleic > palmitic > stearic > ALA > myristic > palmitoleic. All insects contained both essential fatty acids, LA (26.4% to 31.4%) and ALA (0.959% to 9.10%). Overall, the relative concentrations of essential fatty acids are consistent with previously published data, where reported concentrations range from 13.8% to 27.7% for LA and 6.7% to 13.9% for ALA in migratory locust; 16.6% to 33.8% for LA and 0.23% to 1.43% for ALA in yellow mealworm; and 26.26% to 36.67% for LA and 0.7% to 5.13% for ALA in house cricket [51,54,61,62,63,64]. Statistical analyses showed that insect species significantly differ (*p* < 0.05) in most fatty acids, except for arachidic, 11-eicosenoic acid and lignoceric acid (Appendix A).

All investigated insect species demonstrated favorable PUFA/SFA ratios, ranging from 1.13 to 1.16 for yellow mealworm, 0.935 to 0.993 for house cricket, and 1.14 to 1.23 for migratory locust. These values are consistent with previous reports for edible insects [43,51], where PUFA/SFA close to 1 is considered beneficial for cardiovascular health due to their role in improving plasma lipid profiles and reducing low-density lipoprotein (LDL) cholesterol levels. At the same time, a ratio < 0.33 can have an atherogenic effect [65,66,67]. Among the species examined, migratory locust exhibited the most optimal n-6/n-3 fatty acid ratio (3.14–3.29), which falls within the range (<4) for lowering chronic disease risk [68]. In contrast, yellow mealworm and house cricket displayed considerably higher ratios (26.6–27.9 and 17.1–17.5, respectively), reflecting a predominance of n-6 fatty acids that has also been reported for these species in earlier studies [51,64,69].

All insects presented a favorable low index of atherogenicity (IA < 0.5), with values of 0.414–0.417 for yellow mealworm, 0.390–0.417 for house cricket, and 0.367–0.399 for migratory locust. However, only the yellow mealworm demonstrated a thrombogenicity index (IT) below 0.5 (0.342–0.343), compared to house cricket (0.601–0.631) and migratory locust (0.509–0.542). Lower values of IA (<0.5) and IT (<0.5) are associated with a reduced risk of developing cardiovascular disease [49,70]. Finally, all species exhibited a desirable hypocholesterolemic/hypercholesterolemic ratio (h/H > 1), with yellow mealworm showing the highest values (3.63–3.68), followed by migratory locust (2.92–3.16) and house cricket (2.51–2.68). A higher h/H ratio is associated with a more favorable lipid profile, as hypocholesterolemic fatty acids (PUFA and MUFA) promote the reduction of serum cholesterol, whereas hypercholesterolemic fatty acids (mainly C12:0, C14:0, C16:0) elevate it [49,71].

Interestingly, despite successful fermentation, the fatty acid profile of the insect species remained unaffected compared to unfermented insects, regardless of the insect species or fermentation culture used. This finding contrasts with those of other previous studies, although those studies involved different insect species and fermentation cultures. For instance, Liu et al. (2022) fermented black soldier fly larvae for four days using *Bacillus subtilis* and *Aspergillus niger*, resulting in a 25-fold increase in free short-chain fatty acids post-fermentation [39]. Similarly, Hadj Saadoun et al. (2020) fermented prepupae, puparia, and dead adults of black soldier flies with lactic acid bacteria (*Lacticaseibacillus rhamnosus*, *Lactiplantibacillus plantarum*) for 72 h, noting a reduction in typical fatty acids like lauric acid (C12:0) and an increase in minor fatty acids such as short-chain fatty acids, odd-chain fatty acids, and branched-chain fatty acids, which are characteristic of bacterial cell walls [38]. Vasilica et al. (2022) fermented insect flour for 48 h using *L. plantarum* and spontaneous fermentation, observing that controlled fermentation enriched SFA and MUFA, while PUFA decreased during fermentation [43].

Several possible explanations exist for the lack of change in the fatty acid profiles observed in this study. First, the relatively short fermentation duration (48 h) may not have been sufficient for microbial activity to alter lipid components significantly. Second, the microbial strains employed—*L. plantarum* and a commercial starter culture composed of *Staphylococcus xylosus*, *Staphylococcus carnosus*, *Lactobacillus sakei*, *Pediococcus pentosaceus*, and *Debaryomyces hansenii*—are primarily adapted to metabolize carbohydrates and amino acids, rather than lipids [72,73,74]. Consequently, their enzymatic potential for lipid transformation is limited. Third, the processing of insect substrates prior to fermentation could have influenced microbial access and activity. For example, Aleknavičius et al. (2022) reported higher microbial abundance and metabolic activity in non-processed fermented insects compared to processed forms, which may enhance interactions with lipid components [75].

### 3.3. Amino Acid Composition

Taking into account that arginine, cystine and tryptophan were not determined in this study, the total amino acid content (Table 3) ranged from 38.3 g/100 g DW (MLM) to 50.6 g/100 g DW (MLP), while total essential amino acid content ranged from 19.6 g/100 g DW (MLM) to 24.4 g/100 g DW (YM). Essential to total amino acids (E/T) ranged from 44.1% (MLP) to 54.6% (YMP), while essential amino acid index (EAAI, Table 4) ranged from 150 (MLP) to 184 (YMP). Analyzed insects were the highest in valine (3.81−4.58 g/100 g DW), alanine (3.38−4.71 g/100 g DW), glutamic acid (2.74−8.09 g/100 g DW), leucine (3.61−4.23 g/100 g DW), aspartic acid (2.95−4.66 g/100 g DW), but lower in methionine (0.478−0.774 g/100 g DW), ornithine (<0.250−3.20 g/100 g DW) and histidine (0.973−1.64 g/100 g DW). Results align with previously reported values for yellow mealworm, house cricket and migratory locust [51,54,63,64,76].

According to amino acid scores calculated based on reference protein composition for older children (≥3 years) and adults [50], methionine is limiting amino acid for all unfermented and fermented insects, which is in line with previous studies [77,78], while some studies recognized cystine together with methionine [53,62] and tryptophan [76] as limiting amino acids in insect species analyzed in this study. Also, nitrogen to protein conversion factor (Kp) was determined for yellow mealworm (5.41), yellow mealworm fermented with *L. plantarum* (5.06), house cricket (5.05) and house cricket fermented with *L. plantarum* (4.67), which was comparable to Kp factors for yellow mealworm (4.64–7.35) and house cricket (4.17–5.71) reported in previous studies [51,56,62,63,64,79,80,81,82,83,84,85,86]. When calculating Kp, it was taken into account that unmeasured amino acids have an 8% contribution to total amino acids [51], and ornithine was excluded from the total amino acids.

Species differed significantly (*p* < 0.05, Appendix A) in their alanine, aspartic acid, histidine, isoleucine, leucine, lysine, methionine, ornithine, phenylalanine, proline, serine, tyrosine and valine, while total essential and total amino acids were similar. The reasons for these differences could be attributed to various factors such as species-specific metabolism, dietary differences, environmental factors (temperature, humidity, habitat) and adaptations to specific ecological roles or functions (flight, reproduction, or survival mechanisms) as suggested in previous studies [87,88,89].

The observed changes in amino acid profiles following fermentation varied significantly across insect species and microbial cultures. In the case of the yellow mealworm, fermentation with both *L. plantarum* and the commercial starter culture resulted in comparable trends, i.e., an increase in alanine (14–15%) and ornithine (115–117%), but reductions in several key amino acids, including aspartic acid (18–25%), glutamic acid (39–41%), lysine (28–30%), serine (24–25%), threonine (21–23%) and tyrosine (19–20%). These results contrast with those of An et al. (2019), who detected an increase in aspartic and glutamic acid when yellow mealworm was fermented with *L. plantarum* [34].

In the case of the house cricket, both fermentation with *L. plantarum* and the commercial starter culture produced similar changes in the amino acid profile. An increase was observed only in ornithine content (115–117%), while several essential and non-essential amino acids decreased. Specifically, reductions were noted for aspartic acid (8–10%), histidine (0–15%), lysine (10–21%), serine (10–12%), threonine (9–10%), and tyrosine (19–24%). Additionally, the total essential amino acid content decreased by 6–10%. These results suggest a potentially adverse impact on protein quality and overall nutritional value, particularly due to the loss of essential amino acids. These results contradict the findings of Kittibunchakul et al. (2023) and Vasilica et al. (2022), who reported exclusively an increase in amino acids after fermentation with lactic acid bacteria and *L. plantarum* [42,43]. Kittibunchakul et al. (2023) mimicked the process of making shrimp paste Kapi, which also involves salting the substrate and a longer fermentation time, which could be the reason for the differences compared to this study, while Vasilica et al. (2022) used the same fermentation time (48 h) and the same *L. plantarum* culture as in this study.

In the case of the migratory locust, the effects of fermentation with *L. plantarum* and the commercial starter culture differed considerably. Fermentation with *L. plantarum* resulted in increased levels of glutamic acid (123%), histidine (22%), isoleucine (11%), ornithine (1181%), and proline (28%), as well as an overall rise in total amino acid content (13%). However, this treatment also led to reductions in aspartic acid (10%), lysine (15%), serine (28%), threonine (14%), tyrosine (43%), and the essential-to-total amino acid (E/T) ratio (18%). In contrast, fermentation with the commercial starter culture resulted in a substantial increase only in ornithine (359%), while most other amino acids declined in concentration. Notable reductions were observed for aspartic acid (27%), histidine (13%), leucine (11%), lysine (21%), methionine (29%), phenylalanine (19%), proline (11%), serine (44%), threonine (40%), tyrosine (31%), as well as total essential amino acids (19%) and total amino acid content (14%). Overall, the results from both fermentation treatments suggest a potentially adverse impact on protein quality and nutritional value, largely due to the depletion of multiple essential amino acids.

As observed in all three insect species, ornithine concentrations significantly increased following fermentation with both microbial cultures. A potential explanation for this could be the involvement of the arginine deiminase (ADI) pathway, which is strongly associated with lactic acid bacteria [90]. In this pathway, arginine is initially converted to citrulline through the action of the enzyme arginine deiminase. Subsequently, citrulline is converted to ornithine by the enzyme ornithine carbamoyl transferase. This pathway plays a crucial role in bacterial survival, particularly under acidic or nitrogen-limited conditions, and also contributes to adenosine triphosphate (ATP) production through the conversion of arginine, with ammonia and carbon dioxide as by-products [91].

Overall, the nutritional consequences of these shifts are mixed. Although increases in amino acids like ornithine are beneficial, the reduction of essential amino acids, such as lysine, methionine, and threonine, is more concerning. Lysine, methionine + cystine, threonine and tryptophan dominate as limiting amino acids in plant-based diets [92], so their reduction could impair protein quality if insect meals are used in food or feed applications. The results also underscore the importance of strain selection and species compatibility in fermentation design to avoid the degradation of nutritionally important amino acids.

### 3.4. Elemental Composition

Elemental concentrations of fermented and unfermented insects are presented in Table 5, Table 6 and Table 7 and daily intake values for 100 g of fresh insects for adults in Table 8. The elemental concentrations of unfermented insects are in agreement with data obtained by Finke (2002, 2015), Machado et al. (2024), and Vehar et al. (2025), indicating that species differ significantly (*p* < 0.05, Appendix A) in most elements, except for Na, Ni, and Zn [51,62,63,93]. Species-specific patterns in elemental composition have been reported before and are attributed to growing environments (geographic and ecological factors) and processing mechanisms [51,88,94]. However, elements are mainly obtained from dietary sources.

Due to the small number of samples, differences in elemental concentrations were not statistically evaluated; instead, a 10% threshold value was used to confirm the presence of differences, which depended on the insect species and the fermentation culture used. In the case of the yellow mealworm, fermentations with *L. plantarum* and commercial starter culture resulted in increased elemental concentrations. Fermentation with *L. plantarum* increased Al (61%), V (29%), Ni (27%), Ag (34%), Cd (16%), Sn (26%), Sb (14%), Cs (29%), Hg (15%) and U (19%), while commercial starter culture increased Mn (96%), Ag (85%), Cd (13%), Cs (28%), Hg (14%) and U (11%). In case of house cricket, *L. plantarum* increased B (37%) and Ag (14%) and decreased Al (31%), V (16%), Co (14%), Zn (14%), Cd (12%), Hg (20%) and Pb (21%), while commercial starter culture increased B (18%), Mg (12%), Mn (42%), Ni (11%), Sb (19%) and decreased Al (29%), Zn (12%), Cd (15%) and Pb (16%). In the case of the migratory locust, both fermentation cultures mainly increased the concentrations of elements. *L. plantarum* increased Na (20%), Mn (36%), Fe (23%) and decreased Se (15%), while commercial starter culture increased Na (49%), Mg (23%), P (22%), K (25%), Mn (378%), Fe (19%), Co (20%), Zn (12%), Rb (12%) and Sn (25%) and decreased Cu (16%) and Ag (19%). In general, fermentation resulted in increased concentrations of essential elements Fe (19–23%), K (25%), Mg (12–23%), Mn (36–378%), Na (20–49%), P (22%), decreased Se (24%) and Cu (16%), while Zn increased in migratory locust fermented with commercial starter culture (12%) and decreased in house cricket fermented with both cultures (12–14%).

Vasilica et al. (2022) examined essential elements, Cr and Ni, in fermented house cricket using atomic absorption spectrophotometry after ash recovery in 20% HCl [43]. They found that *L. plantarum* increased all elemental concentrations after 24 h, while spontaneous sourdough fermentation increased all except P and Zn, which is only partially consistent with our findings. The authors attributed the increase to a reduction in pH, which may activate phytase and reduce mineral-binding compounds, such as phytates and tannins [95]. In contrast, the present study measured total elemental content after acid digestion with ICP-MS. Therefore, the observed increases likely reflect relative concentration effects, such as a reduction in dry matter during fermentation [96], which concentrates remaining minerals, as well as microbial accumulation or redistribution within the biomass. These effects appear to be species- and culture-dependent.

The results show that consuming 100 g of fermented and unfermented insects in fresh weight (FW) would contribute a substantial portion of the recommended daily intake for adults of Cu (38.1–81.1%), P (40.2–55.8%), and Zn (32.1–63.8%). Specifically, 100 g of fermented and unfermented yellow mealworm would provide 32.0–34.7% of the recommended daily intake of Mg, while house cricket would supply 36.1–53.9% of the requirement for Mn. Daily intake values for all essential elements, except Na and Zn, differ statistically significantly between species (*p* < 0.05, Appendix A). A risk assessment indicates that consuming 100 g of fresh fermented or unfermented insects poses no health risk to adults. Furthermore, based on the maximum levels established by Directive 2002/32/EC of the European Parliament and of the Council of 7 May 2002 on undesirable substances in animal feed for Cu (15–35 mg/kg FW), Fe (30 mg/kg FW), Cd (2 mg/kg FW), Pb (10 mg/kg FW), As (2 mg/kg FW) and Hg (0.1 mg/kg FW) in the feedstuff, all tested insects comply with feed safety standards [97].

### 3.5. Oxidation Stability

Numerous studies have explored the impact of fermentation on oxidation stability in various food products, highlighting its potential to enhance the shelf life and stability of lipids and bioactive compounds [98,99,100,101]. However, limited research has been conducted specifically on the oxidation stability of edible insects, particularly under the fermentation conditions used in this study. Given the constraints on sample material, oxidation stability measurements were conducted with a single measurement per condition (Figure 1). Among the three insect species analyzed, yellow mealworm exhibited the highest oxidation stability (induction time = 209 min), followed by migratory locust (78 min), then house cricket (66 min). The effect of fermentation on oxidation stability depended on the species. In yellow mealworm, fermentation notably reduced the induction time by 42% with *L. plantarum* and by 41% with the commercial starter culture. A similar, though less pronounced decrease, was observed for migratory locust, with induction times reduced by 21% and 29%, respectively. In contrast, house cricket exhibited improved oxidation stability after fermentation, with an increase in induction time of 153% with *L. plantarum* and 167% with the commercial starter culture.

These findings suggest that the impact of fermentation on lipid oxidation stability is primarily dependent on the insect species, while the type of microbial culture used plays a comparatively minor role. Similar observations were reported by Borremans, Smets, and Van Campenhout (2020), who noted the onset of secondary lipid oxidation in yellow mealworm during lactic acid fermentation, with further progression over time [102]. This finding aligns with the results of the current study, which found that fermentation reduced the oxidation stability of yellow mealworm. Bernardo and Conte-Junior (2024) also emphasized that the oxidation stability of edible insects is strongly influenced by species-specific lipid composition and the processing methods used [103]. Insects vary in PUFA content, which contributes to differences in susceptibility to oxidation. In this study, yellow mealworm contained the lowest amount of PUFA (Table 2), potentially explaining its initially higher oxidation stability compared to the other two species, except for fermented house cricket, which showed an unexpected increase in induction time.

Processing steps, such as freezing, freeze-drying, cooking and grinding, can also affect oxidation stability. These treatments may alter the nutritional and physicochemical properties of food products through physical, thermal or chemical mechanisms, consequently decreasing their oxidation stability [104]. In the present study, all insect samples were killed by freezing, freeze-dried, ground, and stored frozen prior to fermentation. These methods are known to increase sample porosity and the specific surface area, making the lipids more vulnerable to oxidation [53,103,105].

Overall, the observed changes in oxidation stability were influenced not only by fermentation but also by prior processing methods. Treatments such as freezing, freeze-drying, and grinding likely contributed to increased lipid exposure and susceptibility to oxidation. Thus, oxidation stability in fermented insect products reflects the combined effects of species-specific lipid composition, fermentation conditions, and processing techniques, all of which should be considered in product development.

## 4. Conclusions

This study demonstrated that the effects of fermenting three edible insect species (yellow mealworm, house cricket, and migratory locust), with *L. plantarum* and a commercial starter culture, are highly dependent on both the insect species and the specific fermentation conditions. Fermentation resulted in a decrease in total dietary fiber and an increase in non-protein nitrogen in both yellow mealworm and house cricket, while other parameters, such as total nitrogen, protein nitrogen, and total fat, remained largely unaffected. No notable improvements were observed in fatty acid profiles, likely due to the relatively short fermentation time. Although increases in specific amino acids, such as ornithine, were beneficial, the reduction of essential amino acids, including lysine, methionine, and threonine, which are often limiting in plant-based diets, was less desirable. Fermentation had mixed effects on mineral composition, enhancing the levels of essential elements like Fe, K, Mg, Mn, Na, P, and Zn in some cases, while reducing others, including Zn, Se, and Cu, with changes reflecting relative concentration effects. Oxidation stability also varied, decreasing in yellow mealworm and migratory locust but improving significantly in house cricket.

These findings highlight the complex interplay between insect species, microbial cultures, and processing conditions in determining nutritional and functional outcomes. Given the limited sample size in this study, future research should include larger sample sets and longer fermentation periods to understand better the mechanisms involved and optimize fermentation strategies for edible insect products. Additionally, it would be valuable to assess bioaccessibility and elemental speciation in future studies to explore how these factors influence the nutritional bioavailability of key micronutrients in fermented insect products.

## Figures and Tables

**Figure 1 foods-14-02929-f001:**
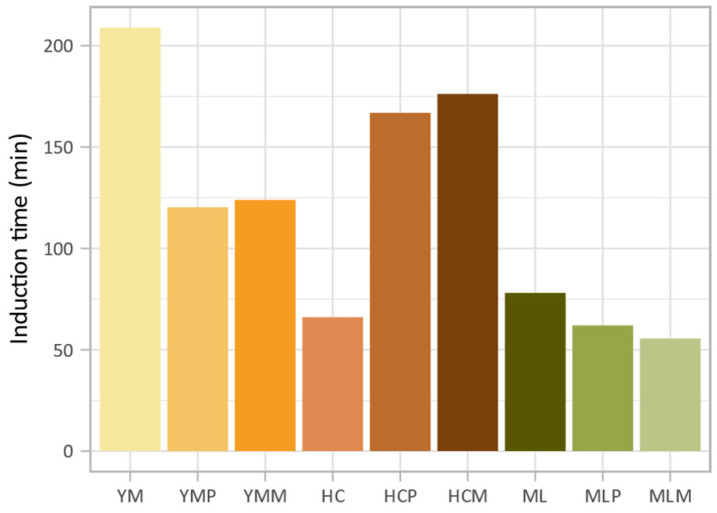
Induction times for insects before and after fermentation (YM: yellow mealworm, YMP: YM + *L. plantarum*, YMM: YM + commercial starter culture, HC: house cricket, HCP: HC + *L. plantarum*, HCM: HC + commercial starter culture, ML: migratory locust, MLP: ML + *L. plantarum*, MLM: ML + commercial starter culture).

**Table 1 foods-14-02929-t001:** Proximate composition of unfermented and fermented (48 h) yellow mealworm and house cricket (lyophilized).

	YM	YMP	HC	HCP
Dry matter [g/100 g]	97.8 ± 0.1	96.3 ± 0.1	97.9 ± 0.1	97.3 ± 0.0
Water [g/100 g]	2.20 ± 0.06	3.70 ± 0.08	2.06 ± 0.08	2.68 ± 0.02
Ash [g/100 g]	4.35 ± 0.02	4.50 ± 0.01	4.30 ± 0.04	4.23 ± 0.02
Total fat [g/100 g]	28.2 ± 1.9	24.4 ± 2.3	22.9 ± 4.6	23.6 ± 2.7
Insoluble dietary fiber [g/100 g]	7.05 ± 0.35	4.38 ± 0.10	6.19 ± 0.53	5.22 ± 0.30
Soluble dietary fiber [g/100 g]	0.025 ± 0.029	0.366 ± 0.259	0.178 ± 0.198	0.366 ± 0.290
Total dietary fiber [g/100 g]	7.08 ± 0.33	4.75 ± 0.35	6.37 ± 0.35	5.59 ± 0.05
Total nitrogen [g/100 g]	9.15 ± 0.03	8.95 ± 0.01	9.24 ± 0.01	9.30 ± 0.03
Non-protein nitrogen [g/100 g]	0.061 ± 0.002	0.196 ± 0.005	0.069 ± 0.001	0.179 ± 0.002
Protein nitrogen [g/100 g]	9.09 ± 0.02	8.75 ± 0.02	9.17 ± 0.01	9.12 ± 0.03
Proteins (6.25) [g/100 g]	56.8 ± 0.1	54.7 ± 0.1	57.3 ± 0.1	57.0 ± 0.2
Available carbohydrates [g/100 g]	0.98 ± 1.90	6.71 ± 2.37	6.64 ± 4.62	5.85 ± 2.66
Energy value [kJ/100 g]	2082 ± 3	1985 ± 3	1985 ± 7	1984 ± 4

YM—yellow mealworm, YMP—yellow mealworm fermented using *L. plantarum*, HC—house cricket, HCP—house cricket fermented using *L. plantarum.*

**Table 2 foods-14-02929-t002:** Contents of the main fatty acids (>1%) saturated fatty acids (SFA), monounsaturated fatty acids (MUFA), polyunsaturated fatty acids (PUFA), n-6 fatty acids, n-3 fatty acids, n-6/n-3 ratio, index of atherogenicity (IA), index of thrombogenicity (IT), hypocholesterolemic/hypercholesterolemic ratio (h/H) in unfermented and fermented insects.

	YM	YMP	YMM	HC	HCP	HCM	ML	MLP	MLM
Myristic acid [%]	3.79 ± 0.02	3.76 ± 0.05	3.76 ± 0.01	0.504 ± 0.005	0.533 ± 0.006	0.525 ± 0.003	1.29 ± 0.01	1.41 ± 0.00	1.41 ± 0.01
Palmitic acid [%]	16.0 ± 0.0	16.2 ± 0.2	16.2 ± 0.1	23.8 ± 0.2	25.0 ± 0.1	24.7 ± 0.2	19.8 ± 0.1	20.9 ± 0.0	20.9 ± 0.1
Palmitoleic acid [%]	1.68 ± 0.00	1.67 ± 0.00	1.67 ± 0.00	0.763 ± 0.013	0.795 ± 0.003	0.794 ± 0.028	0.805 ± 0.013	0.801 ± 0.002	0.813 ± 0.019
Stearic acid [%]	3.20 ± 0.00	3.35 ± 0.03	3.34 ± 0.01	8.61 ± 0.01	8.81 ± 0.02	8.81 ± 0.09	9.59 ± 0.03	9.83 ± 0.04	9.87 ± 0.04
Oleic acid [%]	46.5 ± 0.0	46.0 ± 0.3	46.0 ± 0.1	31.9 ± 0.5	31.4 ± 0.0	31.6 ± 0.1	28.2 ± 0.3	28.0 ± 0.0	27.9 ± 0.0
Linoleic acid [%]	26.4 ± 0.0	26.5 ± 0.0	26.5 ± 0.0	31.4 ± 0.1	30.8 ± 0.0	30.9 ± 0.2	29.8 ± 0.2	28.4 ± 0.0	28.5 ± 0.0
α-linolenic acid [%]	1.00 ± 0.01	0.959 ± 0.002	0.968 ± 0.000	1.85 ± 0.01	1.77 ± 0.01	1.77 ± 0.01	9.06 ± 0.06	9.06 ± 0.00	9.10 ± 0.00
SFA [%]	23.9 ± 0.0	24.4 ± 0.3	24.4 ± 0.1	33.6 ± 0.4	34.9 ± 0.0	34.6 ± 0.3	31.6 ± 0.1	33.1 ± 0.1	33.1 ± 0.1
MUFA [%]	48.5 ± 0.0	47.9 ± 0.3	48.0 ± 0.1	32.9 ± 0.5	32.4 ± 0.0	32.6 ± 0.1	29.5 ± 0.2	29.4 ± 0.0	29.2 ± 0.1
PUFA [%]	27.6 ± 0.0	27.7 ± 0.0	27.7 ± 0.0	33.4 ± 0.1	32.6 ± 0.0	32.8 ± 0.2	38.9 ± 0.2	37.5 ± 0.0	37.7 ± 0.1
PUFA/SFA	1.16 ± 0.00	1.13 ± 0.01	1.14 ± 0.00	0.993 ± 0.009	0.935 ± 0.002	0.947 ± 0.015	1.23 ± 0.00	1.14 ± 0.00	1.14 ± 0.01
n-6 [%]	26.6 ± 0.0	26.7 ± 0.0	26.7 ± 0.0	31.6 ± 0.1	30.9 ± 0.0	31.0 ± 0.2	29.9 ± 0.2	28.5 ± 0.0	28.6 ± 0.1
n-3 [%]	1.00 ± 0.01	0.959 ± 0.002	0.968 ± 0.000	1.85 ± 0.01	1.77 ± 0.01	1.77 ± 0.01	9.06 ± 0.06	9.06 ± 0.00	9.10 ± 0.00
n-6/n-3	26.6 ± 0.2	27.9 ± 0.1	27.6 ± 0.0	17.1 ± 0.0	17.4 ± 0.1	17.5 ± 0.1	3.29 ± 0.05	3.14 ± 0.00	3.14 ± 0.01
IA	0.414 ± 0.001	0.417 ± 0.007	0.417 ± 0.002	0.390 ± 0.005	0.417 ± 0.002	0.411 ± 0.006	0.367 ± 0.003	0.399 ± 0.000	0.399 ± 0.003
IT	0.342 ± 0.000	0.342 ± 0.004	0.343 ± 0.001	0.601 ± 0.006	0.631 ± 0.003	0.624 ± 0.009	0.509 ± 0.003	0.542 ± 0.001	0.542 ± 0.003
h/H	3.68 ± 0.01	3.63 ± 0.05	3.64 ± 0.02	2.68 ± 0.04	2.51 ± 0.01	2.55 ± 0.04	3.16 ± 0.02	2.92 ± 0.00	2.92 ± 0.02

YM: yellow mealworm, YMP: yellow mealworm fermented using *L. plantarum*, YMM: yellow mealworm fermented using commercial starter culture, HC: house cricket, HCP: house cricket fermented using *L. plantarum*, HCM: house cricket fermented using commercial starter culture, ML: migratory locust, MLP: migratory locust fermented using *L. plantarum*, MLM: migratory locust fermented using commercial starter culture. SFA, MUFA, PUFA, n-6, n-3, IA, IT, h/H are calculated based on all detected fatty acids (Appendix A).

**Table 3 foods-14-02929-t003:** Concentrations of amino acids (g/100 g DW), essential and nonessential amino acid content (g/100 g DW) and essential total amino acids ratio (%) in unfermented and fermented insects.

	YM	YMP	YMM	HC	HCP	HCM	ML	MLP	MLM
Alanine	3.38 ± 0.12	3.90 ± 0.06	3.84 ± 0.02	4.04 ± 0.07	3.98 ± 0.16	4.07 ± 0.15	4.71 ± 0.23	4.64 ± 0.17	4.54 ± 0.56
Aspartic acid	4.66 ± 0.30	3.52 ± 0.10	3.84 ± 0.09	4.34 ± 0.15	3.89 ± 0.08	4.01 ± 0.15	4.01 ± 0.21	3.60 ± 0.14	2.95 ± 0.08
Glutamic acid	4.67 ± 0.48	2.84 ± 0.10	2.74 ± 0.09	4.09 ± 0.54	3.76 ± 0.33	3.88 ± 0.23	3.63 ± 0.39	8.09 ± 0.31	3.30 ± 0.18
Glycine	2.65 ± 0.09	2.82 ± 0.03	2.77 ± 0.04	2.78 ± 0.14	2.66 ± 0.07	2.69 ± 0.01	2.83 ± 0.08	3.08 ± 0.15	2.69 ± 0.09
Histidine	1.57 ± 0.06	1.64 ± 0.02	1.60 ± 0.02	1.14 ± 0.01	0.973 ± 0.018	1.14 ± 0.04	1.20 ± 0.05	1.46 ± 0.05	1.05 ± 0.06
Isoleucine	2.10 ± 0.05	2.25 ± 0.05	2.19 ± 0.02	1.94 ± 0.01	1.92 ± 0.04	1.96 ± 0.04	2.03 ± 0.08	2.25 ± 0.05	1.91 ± 0.07
Leucine	4.12 ± 0.16	4.23 ± 0.09	4.16 ± 0.06	4.03 ± 0.05	3.88 ± 0.08	3.95 ± 0.10	4.05 ± 0.17	4.14 ± 0.08	3.61 ± 0.16
Lysine	3.58 ± 0.21	2.56 ± 0.09	2.51 ± 0.06	3.87 ± 0.06	3.05 ± 0.04	3.49 ± 0.04	3.83 ± 0.17	3.25 ± 0.38	3.01 ± 0.10
Methionine	0.670 ± 0.036	0.677 ± 0.039	0.652 ± 0.003	0.774 ± 0.027	0.742 ± 0.016	0.748 ± 0.026	0.672 ± 0.045	0.660 ± 0.024	0.478 ± 0.020
Ornithine	<0.250	0.537 ± 0.037	0.542 ± 0.112	<0.250	0.699 ± 0.227	0.774 ± 0.234	<0.250	3.20 ± 1.10	1.15 ± 0.24
Phenylalanine	1.97 ± 0.10	1.99 ± 0.02	2.03 ± 0.06	1.93 ± 0.07	1.78 ± 0.03	1.81 ± 0.02	2.04 ± 0.08	1.92 ± 0.20	1.66 ± 0.05
Proline	3.00 ± 0.12	3.20 ± 0.06	3.11 ± 0.04	2.92 ± 0.06	2.76 ± 0.09	2.86 ± 0.10	3.14 ± 0.09	4.03 ± 0.07	2.80 ± 0.16
Serine	2.73 ± 0.08	2.05 ± 0.06	2.09 ± 0.07	2.14 ± 0.33	1.88 ± 0.07	1.93 ± 0.09	2.32 ± 0.10	1.66 ± 0.32	1.31 ± 0.41
Threonine	2.43 ± 0.11	1.88 ± 0.02	1.92 ± 0.03	2.09 ± 0.17	1.90 ± 0.04	1.88 ± 0.08	2.35 ± 0.14	2.02 ± 0.39	1.40 ± 0.45
Tyrosine	3.84 ± 0.13	3.07 ± 0.07	3.12 ± 0.05	3.16 ± 0.06	2.40 ± 0.08	2.56 ± 0.11	3.53 ± 0.02	2.01 ± 0.28	2.42 ± 0.48
Valine	4.11 ± 0.18	4.40 ± 0.07	4.36 ± 0.01	3.90 ± 0.04	3.81 ± 0.09	3.91 ± 0.16	4.38 ± 0.15	4.58 ± 0.06	4.05 ± 0.22
Total EAA	24.4 ± 1.0	22.7 ± 0.3	22.5 ± 0.1	22.8 ± 0.3	20.5 ± 0.4	21.4 ± 0.5	24.1 ± 0.8	22.3 ± 0.9	19.6 ± 0.7
Total AA	45.5 ± 1.5	41.6 ± 0.3	41.5 ± 0.3	43.1 ± 0.9	40.1 ± 0.3	41.7 ± 1.0	44.7 ± 1.2	50.6 ± 1.5	38.3 ± 1.7
E/T [%]	53.6 ± 0.7	54.6 ± 0.4	54.4 ± 0.3	52.9 ± 0.7	51.0 ± 0.7	51.5 ± 0.3	53.8 ± 0.5	44.1 ± 1.6	51.1 ± 0.5

YM: yellow mealworm, YMP: yellow mealworm fermented using *L. plantarum*, YMM: yellow mealworm fermented using commercial starter culture, HC: house cricket, HCP: house cricket fermented using *L. plantarum*, HCM: house cricket fermented using commercial starter culture, ML: migratory locust, MLP: migratory locust fermented using *L. plantarum*, MLM: migratory locust fermented using commercial starter culture.

**Table 4 foods-14-02929-t004:** Amino acid scores and essential amino acid index [%].

	YM	YMP	YMM	HC	HCP	HCM	ML	MLP	MLM
Histidine	2.15 ± 0.07	2.47 ± 0.03	2.41 ± 0.03	1.65 ± 0.02	1.52 ± 0.02	1.70 ± 0.02	1.67 ± 0.07	1.80 ± 0.09	1.71 ± 0.03
Isoleucine	1.54 ± 0.02	1.80 ± 0.02	1.76 ± 0.02	1.50 ± 0.04	1.60 ± 0.02	1.57 ± 0.01	1.51 ± 0.02	1.49 ± 0.05	1.66 ± 0.02
Leucine	1.54 ± 0.01	1.73 ± 0.03	1.70 ± 0.01	1.58 ± 0.02	1.64 ± 0.03	1.61 ± 0.01	1.53 ± 0.02	1.39 ± 0.04	1.59 ± 0.01
Lysine	1.75 ± 0.07	1.37 ± 0.04	1.34 ± 0.04	1.99 ± 0.03	1.69 ± 0.01	1.86 ± 0.05	1.90 ± 0.04	1.43 ± 0.15	1.75 ± 0.02
Methionine	0.669 ± 0.014	0.74 ± 0.04	0.714 ± 0.006	0.816 ± 0.013	0.842 ± 0.018	0.816 ± 0.010	0.682 ± 0.027	0.593 ± 0.015	0.568 ± 0.010
Phenylalanine + Tyrosine	3.36 ± 0.02	3.20 ± 0.06	3.27 ± 0.01	3.11 ± 0.03	2.75 ± 0.04	2.76 ± 0.04	3.28 ± 0.05	2.05 ± 0.13	2.80 ± 0.21
Threonine	2.33 ± 0.08	1.97 ± 0.02	2.01 ± 0.02	2.11 ± 0.16	2.06 ± 0.04	1.96 ± 0.04	2.28 ± 0.07	1.73 ± 0.31	1.60 ± 0.57
Valine	2.32 ± 0.05	2.71 ± 0.04	2.70 ± 0.02	2.32 ± 0.02	2.44 ± 0.04	2.40 ± 0.05	2.51 ± 0.09	2.32 ± 0.05	2.71 ± 0.04
EAAI [%]	179 ± 2	184 ± 2	182 ± 1	177 ± 2	172 ± 2	174 ± 1	177 ± 2	150 ± 6	164 ± 6

YM: yellow mealworm, YMP: yellow mealworm fermented using *L. plantarum*, YMM: yellow mealworm fermented using commercial starter culture, HC: house cricket, HCP: house cricket fermented using *L. plantarum*, HCM: house cricket fermented using commercial starter culture, ML: migratory locust, MLP: migratory locust fermented using *L. plantarum*, MLM: migratory locust fermented using commercial starter culture.

**Table 5 foods-14-02929-t005:** Makro elements (g/kg FW) in unfermented and fermented insects.

Elements	YM	YMP	YMM	HC	HCP	HCM	ML	MLP	MLM
Ca	0.155 ± 0.016	0.148 ± 0.005	0.154 ± 0.004	0.379 ± 0.030	0.368 ± 0.012	0.414 ± 0.002	0.271 ± 0.025	0.277 ± 0.006	0.305 ± 0.033
K	2.71 ± 0.15	2.66 ± 0.00	2.62 ± 0.14	2.90 ± 0.02	2.79 ± 0.03	3.00 ± 0.15	3.13 ± 0.00	3.11 ± 0.23	3.90 ± 0.02
Mg	1.04 ± 0.05	0.960 ± 0.043	1.00 ± 0.05	0.216 ± 0.007	0.213 ± 0.006	0.242 ± 0.015	0.268 ± 0.016	0.272 ± 0.023	0.329 ± 0.015
Na	0.388 ± 0.023	0.383 ± 0.007	0.396 ± 0.025	1.07 ± 0.01	1.04 ± 0.01	1.13 ± 0.02	0.232 ± 0.004	0.279 ± 0.009	0.346 ± 0.030
P	2.79 ± 0.10	2.67 ± 0.06	2.69 ± 0.15	2.14 ± 0.02	2.01 ± 0.00	2.19 ± 0.04	2.07 ± 0.05	2.12 ± 0.21	2.51 ± 0.02
S	1.04 ± 0.07	1.03 ± 0.02	1.02 ± 0.03	1.38 ± 0.01	1.34 ± 0.04	1.39 ± 0.00	1.09 ± 0.04	1.08 ± 0.10	1.14

YM: yellow mealworm, YMP: yellow mealworm fermented using L. plantarum, YMM: yellow mealworm fermented using commercial starter culture, HC: house cricket, HCP: house cricket fermented using L. plantarum, HCM: house cricket fermented using commercial starter culture, ML: migratory locust, MLP: migratory locust fermented using L. plantarum, MLM: migratory locust fermented using commercial starter culture.

**Table 6 foods-14-02929-t006:** Mikro elements (mg/kg FW) in unfermented and fermented insects.

Elements	YM	YMP	YMM	HC	HCP	HCM	ML	MLP	MLM
Al	0.607 ± 0.133	0.803 ± 0.070	0.646 ± 0.052	24.9 ± 4.1	17.2 ± 3.5	17.8 ± 2.9	23.3 ± 8.9	37.6 ± 26.7	27.1 ± 6.6
B	1.11 ± 0.17	1.05 ± 0.06	1.03 ± 0.01	0.257 ± 0.036	0.351 ± 0.024	0.303 ± 0.024	0.337 ± 0.029	0.356 ± 0.001	0.329
Cu	5.07 ± 0.33	4.95 ± 0.03	4.95 ± 0.26	5.96 ± 0.12	5.67 ± 0.11	6.27 ± 0.38	10.0 ± 0.4	10.5 ± 0.4	8.44 ± 0.02
Fe	13.8 ± 1.3	12.9 ± 0.5	12.9 ± 0.5	17.4 ± 0.3	15.7 ± 0.8	18.4 ± 0.3	15.7 ± 0.1	19.3 ± 1.7	18.7 ± 0.0
Mn	2.93 ± 0.25	2.88 ± 0.06	5.75 ± 0.22	11.4 ± 0.2	10.8 ± 0.1	16.2 ± 0.7	0.775 ± 0.048	1.05 ± 0.20	3.70 ± 1.23
Mo	0.332 ± 0.003	0.353 ± 0.010	0.336 ± 0.006	0.249 ± 0.001	0.255 ± 0.000	0.251 ± 0.004	0.311 ± 0.063	0.267 ± 0.001	0.277 ± 0.025
Ni	0.147 ± 0.021	0.187 ± 0.030	0.16 ± 0.00	0.0645 ± 0.0063	0.0743 ± 0.0187	0.0716 ± 0.0012	0.256 ± 0.033	0.235 ± 0.004	0.254 ± 0.014
Rb	0.239 ± 0.014	0.226 ± 0.004	0.237 ± 0.018	0.967 ± 0.019	0.883 ± 0.021	0.975 ± 0.012	0.798 ± 0.001	0.796 ± 0.038	0.986 ± 0.013
Sr	0.542 ± 0.075	0.525 ± 0.010	0.551 ± 0.052	0.590 ± 0.046	0.556 ± 0.058	0.614 ± 0.027	0.697 ± 0.057	0.662 ± 0.023	0.759 ± 0.036
Zn	40.0 ± 3.4	39.3 ± 0.2	40.4 ± 3.3	70.2 ± 5.4	60.2 ± 6.8	61.7 ± 1.2	35.5 ± 0.5	35.4 ± 2.7	39.8 ± 0.3

YM: yellow mealworm, YMP: yellow mealworm fermented using *L. plantarum*, YMM: yellow mealworm fermented using commercial starter culture, HC: house cricket, HCP: house cricket fermented using *L. plantarum*, HCM: house cricket fermented using commercial starter culture, ML: migratory locust, MLP: migratory locust fermented using *L. plantarum*, MLM: migratory locust fermented using commercial starter culture.

**Table 7 foods-14-02929-t007:** Trace elements (µg/kg FW) in unfermented and fermented insects.

Elements	YM	YMP	YMM	HC	HCP	HCM	ML	MLP	MLM
Ag	0.507 ± 0.029	0.681 ± 0.044	0.937 ± 0.116	0.866 ± 0.028	0.989 ± 0.012	0.850 ± 0.202	2.62 ± 0.54	2.71 ± 0.37	2.11 ± 0.04
As	15.4 ± 2.3	16.1 ± 0.6	16.1 ± 0.4	4.81 ± 2.00	5.09 ± 1.32	4.01 ± 0.75	6.97 ± 0.40	7.57 ± 1.22	19.4 ± 18.5
Cd	25.6 ± 0.5	29.8 ± 0.8	29.0 ± 0.4	11.2 ± 1.3	9.82 ± 0.40	9.52 ± 0.37	12.7 ± 8.0	15.6 ± 3.1	11.2 ± 4.4
Co	17.5 ± 1.6	18.0 ± 0.8	17.3 ± 0.1	23.8 ± 0.7	20.4 ± 0.7	21.6 ± 0.8	8.60 ± 1.00	10.6 ± 3.4	10.3 ± 0.4
Cs	0.382 ± 0.016	0.494 ± 0.052	0.490 ± 0.076	4.46 ± 0.04	4.29 ± 0.39	4.53 ± 0.10	3.68 ± 2.87	1.86	4.35 ± 3.25
Hg	0.282 ± 0.033	0.325 ± 0.002	0.321 ± 0.004	0.641 ± 0.042	0.516 ± 0.025	0.617 ± 0.068	1.06 ± 0.23	1.24 ± 0.18	0.970 ± 0.230
Pb	2.16 ± 0.16	2.34 ± 0.10	2.06 ± 0.03	33.2 ± 0.2	26.3 ± 3.2	28.0 ± 1.1	31.4 ± 5.4	43.2 ± 18.0	34.0 ± 2.67
Sb	0.532 ± 0.018	0.607 ± 0.034	0.493 ± 0.034	1.37 ± 0.20	1.41 ± 0.12	1.63 ± 0.16	2.55 ± 0.41	3.37 ± 0.81	2.78 ± 0.58
Se	42.4 ± 0.6	45.4 ± 0.1	43.6 ± 1.7	98.0 ± 2.3	99.7 ± 4.6	95.1 ± 6.8	83.7 ± 10.1	71.0 ± 5.70	88.9 ± 1.1
Sn	6.13 ± 1.29	7.72 ± 0.09	7.70 ± 1.42	15.7 ± 2.1	14.9 ± 2.9	13.6 ± 1.2	10.1 ± 0.4	12.9 ± 4.0	12.6 ± 2.9
U	23.6 ± 0.4	28.2 ± 1.4	26.2 ± 0.5	31.6 ± 8.4	28.4 ± 2.7	27.8 ± 0.6	1.29 ± 0.21	1.55 ± 0.45	1.21 ± 0.23
V	52.9 ± 7.5	61.6 ± 0.0	58.9 ± 1.1	63.8 ± 9.1	53.9 ± 1.6	56.7 ± 0.7	18.2 ± 4.8	23.5 ± 6.0	17.7 ± 0.7

YM: yellow mealworm, YMP: yellow mealworm fermented using *L. plantarum*, YMM: yellow mealworm fermented using commercial starter culture, HC: house cricket, HCP: house cricket fermented using *L. plantarum*, HCM: house cricket fermented using commercial starter culture, ML: migratory locust, MLP: migratory locust fermented using *L. plantarum*, MLM: migratory locust fermented using commercial starter culture.

**Table 8 foods-14-02929-t008:** Daily intake values (%) for 100 g of fresh insects, calculated for adults.

	RDI (mg/day)	YM	YMP	YMM	HC	HCP	HCM	ML	MLP	MLM
Ca	1000	1.5 ± 0.2	1.5 ± 0.1	1.5 ± 0.0	3.8 ± 0.3	3.7 ± 0.1	4.1 ± 0.0	2.7 ± 0.3	2.8 ± 0.1	3.1 ± 0.3
Cu	1.3	39.0 ± 2.6	38.1 ± 0.2	38.1 ± 2.0	45.9 ± 0.9	43.6 ± 0.8	48.3 ± 2.9	76.9 ± 3.0	81.1 ± 2.9	64.9 ± 0.1
Fe	16	8.6 ± 0.8	8.1 ± 0.3	8.1 ± 0.5	10.8 ± 0.2	9.8 ± 0.5	11.5 ± 0.2	9.8 ± 0.0	12.1 ± 1.1	11.7 ± 0.0
K	3500	7.7 ± 0.4	7.6 ± 0.0	7.5 ± 0.4	8.3 ± 0.0	8.0 ± 0.1	8.6 ± 0.4	8.9 ± 0.0	8.9 ± 0.7	11.2 ± 0.1
Mg	300	34.7 ± 1.7	32.0 ± 1.4	33.5 ± 1.6	7.2 ± 0.2	7.1 ± 0.2	8.1 ± 0.5	8.9 ± 0.6	9.1 ± 0.8	11.0 ± 0.5
Mn	3	9.8 ± 0.8	9.6 ± 0.2	19.2 ± 0.7	38.0 ± 0.6	36.1 ± 0.3	53.9 ± 2.3	2.6 ± 0.2	3.5 ± 0.7	12.3 ± 4.1
Na	2000	1.9 ± 0.1	1.9 ± 0.0	2.0 ± 0.1	5.4 ± 0.0	5.2 ± 0.0	5.7 ± 0.1	1.2 ± 0.0	1.4 ± 0.0	1.7 ± 0.1
P	500	55.8 ± 2.0	53.4 ± 1.2	53.8 ± 3.0	42.8 ± 0.4	40.2 ± 0.1	43.8 ± 0.8	41.4 ± 1.1	42.3 ± 4.1	50.3 ± 0.5
Se	0.07	6.1 ± 0.1	6.5 ± 0.0	6.2 ± 0.2	14.0 ± 0.3	14.2 ± 0.7	13.6 ± 1.0	12.0 ± 1.4	10.1 ± 0.8	12.7 ± 0.2
Zn	11	36.3 ± 3.1	35.7 ± 0.2	36.7 ± 3	63.8 ± 4.9	54.7 ± 6.2	56.1 ± 1.1	32.3 ± 0.4	32.1 ± 2.5	36.2 ± 0.3

YM: yellow mealworm, YMP: yellow mealworm fermented using *L. plantarum*, YMM: yellow mealworm fermented using commercial starter culture, HC: house cricket, HCP: house cricket fermented using *L. plantarum*, HCM: house cricket fermented using commercial starter culture, ML: migratory locust, MLP: migratory locust fermented using *L. plantarum*, MLM: migratory locust fermented using commercial starter culture.

## Data Availability

Data will be made available on request.

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
