# Peer review of "Evaluation of Nutritional Quality and Oxidation Stability of Fermented Edible Insects"

_foods, 2025, doi:10.3390/foods14172929_

Round 1

Reviewer 1 Report

Comments and Suggestions for Authors

The topic is interesting as insect proteins are rich in essential amino acids. A lot of typographical mistakes were observed. So, make corrections accordingly:

Line No 21: Ensure that throughout the entire manuscript, percentage ranges are written as '21–29%' instead of '21%–29%' to maintain consistency and avoid repeating the percentage sign and any other sign you used in the article.

In your graphical abstract: "The graphical abstract is incomplete. It lacks data analysis, materials, and methods, and does not comprehensively represent the study. A graphical abstract should include all key information covered in the manuscript to provide a complete overview of the research."

Please redraw the graphical abstract to include all essential components such as materials and methods, data analysis, and key findings so that it accurately reflects the entire manuscript.

In the introduction section, the introduction should include sufficient background, a relevant literature review, a clear problem statement, study objectives, and highlight the novelty and significance of the work. Ensure it covers the importance of fortified foods, the role of Vitamin D and calcium, the relevance of multi-millet extruded snacks, existing research gaps, and the objectives of the present study.

In the material and method: Line no 128: Please explain and justify how 30% humidity was maintained during the experiment. Include details of the equipment or method used to control and monitor humidity to ensure reproducibility.

Line no 143: Freeze killing, explain and elaborate.

Line No 155: Overnight, be specific, write the exact hour.

Line No 168: Please specify the operating conditions of the lyophilizer (e.g., temperature, pressure, and drying time) used in the experiment to ensure reproducibility and clarity.

Line no 169: Please ensure consistency in the use of time units. For example, 'h' is used in Line 169 and 'hours' in Line 171. Choose one format and make it homogeneous and consistent throughout the manuscript. This symmetry should be maintained across the entire manuscript. Same for Line no 239, 366, 373,

In Section 2.4: This section is too lengthy for proximate composition. There is no need to elaborate so much; keep it concise, mention only the exact analysis names (e.g., carbohydrate, protein, fat, ash, fiber), and include the missing formulas where necessary.

Section 3: Results: Please correct '(Error! Reference source not found.)' with the appropriate table or figure number. Such errors are not acceptable in the final manuscript.

Please ensure that statistical interpretations are included in the discussion. The manuscript currently lacks statements indicating whether differences were statistically significant or non-significant (e.g., p > 0.05) for the results presented.

There is no need to mention standard deviation (SD) values in the discussion section. Focus on interpreting the results rather than repeating statistical details, which are better suited for tables or the results section. E.g., Lino no: 360

Lino no 367: A 1364% increase in soluble dietary fiber content after 48 h fermentation appears highly unrealistic. Please recheck the data, verify the calculation method, and provide strong justification or supporting references. Without validation, such an extreme change cannot be accepted in the manuscript.

Line no 380-381: A 585% increase in available carbohydrates after fermentation seems highly implausible. Please recheck the raw data, calculation method, and units. Provide a clear biochemical explanation or literature support if this value is accurate, as such a large change is not typical for fermentation.

Each table and figure Lacks proper statistical justification. There is no evidence of statistical analysis, superscript lettering, or indication of significant differences among samples. The results section also fails to report any statistical interpretation. This is a critical issue and must be addressed before the manuscript can be considered for publication.

Line no 422: Don't need to mention SD value in Discussion. Please incorporate this comment in the whole manuscript.

Line no 440-454: In the results and discussion section, numeric values are highlighted, but there is insufficient interpretation and scientific reasoning provided. Please expand the discussion with strong validation by linking the findings to underlying mechanisms, relevant literature, and comparisons with previous studies rather than only stating numbers. It should be reflected in the whole manuscript.

In Section 3.4, two different tables are both labelled as Table 4, which creates confusion. Please correct the numbering to ensure each table has a unique identifier and is referenced properly in the text.

Similar issues are present with Tables 2 and 3, where duplicate numbering or mislabelling creates confusion. Please review all tables carefully and correct the numbering to ensure each table is uniquely identified and properly referenced in the manuscript."

In Table 4 under elemental analysis, you have mentioned RDI values; however, the Recommended Daily Intake varies with age groups. Please provide age-wise RDI data or specify the reference group (e.g., adults, children) to make the results accurate and scientifically valid.

Line no 678: The phrase '(Error! Reference source not found.)' indicates a broken cross-reference and must be corrected or removed. Additionally, when discussing oxidative stability, acknowledge that there are numerous papers available on the topic while emphasizing that limited research has been conducted in the specific context of your study. Revise the section accordingly to reflect both the extensive existing literature and the research gap.

Please avoid claiming that the analysis was not performed in replicates due to insufficient sample. Such statements weaken the scientific rigor of the study. If replicates were not conducted, provide a valid methodological or experimental justification instead.

Table 4 lacks standard deviation (SD) values and any indication of statistical analysis. Please include SDs for all measured parameters and apply appropriate statistical tests with superscripts or annotations to show significant differences among samples.

The table appears ambiguous as the decimal representation is inconsistent; some values are shown with three decimal places, while others have two or fewer. Please standardize the number of decimal places across all values to maintain uniformity and clarity.

The legend for Figure 1 states 'Induction times for insects before and after fermentation,' but the figure does not provide any information distinguishing between before and after fermentation. Please revise the figure and legend to accurately reflect the data presented or include the missing information.

Figure 1 also lacks statistical analysis. Please include appropriate statistical data (e.g., SD values, error bars, and superscripts indicating significant differences) to validate the results and ensure scientific accuracy.

Overall, the manuscript suffers from major scientific and presentation flaws that must be addressed before it can be considered for publication.

Reviewer 2 Report

Comments and Suggestions for Authors

In this paper the Evaluation of Nutritional Quality and Oxidation Stability in Fermented Edible Insects are described.  Your claim of originality is that “differences in insect species, fermentation conditions, and analytical methods limit cross-study comparisons and highlight a lack of standardized, comparative data across multiple insect types”. You mention several studies on fermented edible insects in the Introduction except your recent work

Jamnik, P., Mahnič, N., Ekselenski, S., Pogačnik da Silva, L., Čadež, N., Membrino, V., Poklar Ulrih, N., Plateis, Z., Toplak, N., Koren, S., Kulma, M., Kouřimská, L., Škvorová, P., & Jeršek, B. (2025). Microbial and biochemical characterisation of fermented house crickets (Acheta domesticus) and mealworm larvae (Tenebrio molitor). Journal of Insects as Food and Feed (published online ahead of print 2025). https://doi.org/10.1163/23524588-bja10246,

Please elaborate on originality in relation to previous studies (including your study) on same insects, which are cited by you
e.g. Borremans, A.; Lenaerts, S.; Crauwels, S.; Lievens, B.; Van Campenhout, L. Marination and Fermentation of Yel-827 low Mealworm Larvae (Tenebrio Molitor). Food Control 2018, 92, 47–52, https://doi.org/10.1016/j.foodcont.2018.04.036 .
Mendoza-Salazar, A.; Santiago-López, L.; Torres-Llanez, M.J.; Hernández-Mendoza, A.; Vallejo-Cordoba, B.; Liceaga, A.M.; González-Córdova, A.F. In Vitro Antioxidant and Antihypertensive Activity of Edible Insects Flours (Mealworm and Grasshopper) Fermented with Lactococcus lactis Strains. Fermentation 2021, 7, 153. https://doi.org/10.3390/fermentation7030153

 Materials and methods

Insect fermentation L 154.: Why did you use only 2 ml of a 20 ml culture?

2.4. Determination of Proximate Composition L. 184 please state the original number of insects and the average weight of each insect

  1. 191 state gravimetric soxhlet , solvents and ratios, which appear under fatty acid analysis
  2. 193 Kjeldahl: A sample mass between 0.5 to 2.0 grams of homogeneous, air-dried samples is required. That amount corresponds to a number of insects which is not mentioned. State number of larvae and mean weight of each.
  3. 197 De Vries method: this method has been validated for milk samples. Have you validated the method for your samples?

2.5. Fatty acid analysis.

  1. how many insects per sample species? I am afraid that your sample is too small
  2. why did you not use the Soxhlet product?
  3. 256. A small sample may lead to great variation

Results

 Table 1: Please check: sums should be 100

Table 2:

please check your sums of SFAs, MUFA and PUFA,  n and saturation for each acid eg C14:0, C18:1etc and the n3 and which the n6

3.5. Oxidation stability: Have you determined TBARS with and without fermentation? Excessive oxidation can lead to the formation of harmful compounds that contribute to chronic diseases

Discussion 622

The levels of cadmium appear quite high!! EFSA has established a TWI for cadmium of 2.5 micrograms per kilogram of body weight.

Also Cesium, mercury and uranium are not essential nutrients for plants or humans and can be harmful at high concentrations. 

Also where do you attribute the element levels results?

Round 2

Reviewer 1 Report

Comments and Suggestions for Authors

Tables 1–8: Please perform ANOVA to identify and indicate the statistically significant differences among the contents of the various samples. This will help strengthen the interpretation of the results and improve the clarity of the tables.

Reviewer 2 Report

Comments and Suggestions for Authors

Although points raised were not addressed, a significant effort was made to improve the MS. 

Author Response

Reviewer #2 did not provide specific suggestions. We nevertheless appreciate their time and effort in evaluating our manuscript.